# The Use of Administrative Data to Investigate the Population Burden of Hepatic Encephalopathy

**DOI:** 10.3390/jcm9113620

**Published:** 2020-11-10

**Authors:** Patricia P. Bloom, Elliot B. Tapper

**Affiliations:** 1Division of Gastroenterology and Hepatology, University of Michigan, Ann Arbor, MI 48109, USA; etapper@umich.edu; 2Gastroenterology Section, VA Ann Arbor Healthcare System, Ann Arbor, MI 48109, USA

**Keywords:** cirrhosis, liver disease, epidemiology

## Abstract

Hepatic encephalopathy (HE) is a devastating complication of cirrhosis with an increasing footprint in global public health. Although the condition is defined using a careful history and examination, we cannot accurately measure the true impact of HE relying on data collected exclusively from clinical studies. For this reason, administrative data sources are necessary to study the population burden of HE. Administrative data is generated with each health care encounter to account for health care resource utilization and is extracted into a dataset for the secondary purpose of research. In order to utilize such data for valid analysis, several pitfalls must be avoided—specifically, selecting the particular database capable of meeting the needs of the study’s aims, paying careful attention to the limits of each given database, and ensuring validity of case definition for HE specific to the dataset. In this review, we summarize the types of data available for and the results of administrative data studies of HE.

## 1. Introduction

Cirrhosis is an increasingly common [1], morbid, and deadly condition [2]. The increased health care utilization [3], symptom burden [4], and mortality associated with cirrhosis is particularly driven by the development of hepatic encephalopathy (HE) [5]. HE is a syndrome of brain dysfunction caused by liver insufficiency and/or portal-systemic shunting that manifests as a spectrum of neuropsychiatric perturbations ranging from deficits in executive functioning to coma [6]. As such, HE is a clinical diagnosis best made in conjunction with a careful clinical examination and exclusion of other causes of altered mentation. Research on the burden and impact of HE at the population level is therefore challenging.

One solution is the use of administrative data. Such data is generated with each health care encounter to account for health care resource utilization and can be extracted into a dataset for the secondary purpose of research. The richness of the included variables, and therefore the questions for which a dataset is amenable, varies with the purpose of primary data collection. At a minimum, administrative databases include demographics and diagnosis or procedure codes (e.g., ICD-10) which are input by clinicians or staff for billing or resource monitoring. The contents of administrative data are only as valid as the methods used to record the clinical details. Administrative data cannot provide the accuracy and granularity of detail found in well-executed prospective clinical research. However, administrative data offers several advantages.

Administrative data allows an understanding of the impact of HE on the population. The careful, prospective, multicenter data needed to define the incidence, health care utilization, and clinical outcomes associated with HE has prohibitive costs. As such, the ability to extract insights from data recorded for other purposes is essential to extend our knowledge of HE epidemiology. To ensure validity, this requires its own deliberate methodology. Herein, we review the tools required to analyze and what is known about HE from administrative sources.

## 2. Identifying Cirrhosis with Administrative Data

Cohort studies using administrative data to identify patients pose unique challenges to investigators wishing to communicate their results. Whereas prospective studies define cirrhosis using clinical criteria with prima facie validity such as histology or clinical criteria supported by imaging and laboratory evidence with an acceptable, largely unquestioned degree of uncertainty, administrative data lacks the assumption of validity. When clinicians or administrative staff process visit charges, they assign billing diagnoses utilizing a system known as the International Classification of Diseases (ICD). The ICD systems and codes utilized vary across time and locality. Whereas much of Europe has used the 10th iteration (ICD-10) for decades, the United States switched from ICD-9 to ICD-10 in October of 2015. These codes may be chosen incorrectly (reducing specificity) or the chosen codes may incompletely catalogue the patient’s active problems (reducing sensitivity). Further, the temporality of codes can only be inferred. The first appearance of a code is felt to establish the index data for a diagnosis but this may lag. Similarly, the prior use of a code does not establish whether it is persistent, resolved, or entered in error.

The use of administrative data to identify patients is therefore dependent on the validation of the codes utilized. Algorithms for the identification of cirrhosis have been established by a number of investigators for a variety of datasets by using chart review to confirm the positive and negative predictive values of diagnostic coding schema [7,8,9,10,11,12]. In general, most approaches involve requiring a specific set of codes and multiple (>1) entries of the codes in outpatient records (or one entry in inpatient records). The performance of diagnostic codes is also etiology dependent. Performance is best for viral hepatitis, moderate for ALD, and worst for NAFLD [8,13,14].

## 3. Identifying Hepatic Encephalopathy with Administrative Data

Numerous studies have used administrative data to identify patients with HE, but only a few have validated the use of such data (Table 1). Kanwal and colleagues validated the use of the ICD-9 code for hepatic encephalopathy (572.2) in a Veterans Affairs (VA) cohort [15]. They found that the presence of at least one 572.2 code had high positive predictive value (0.86) and high negative predictive value (0.87) for a diagnosis of HE on detailed chart review with the denominator of persons with multiple cirrhosis codes. An algorithm based on the ICD-9 code for HE (572.2) and prescription fills for lactulose or rifaximin had moderate agreement with a chart review diagnosis of HE in a separate VA cohort [16]. Most published studies using administrative data to identify HE have used ICD-9 codes. However, to use United States data after 2015 when ICD-9 was abandoned, algorithms using ICD-10 are needed.

Unfortunately, the ICD-10 system lacks a code for HE. In this vacuum, coders will use a handful of different options. As we have found, across the US this most frequently this involves the code K72.90, which is technically “hepatic failure, not otherwise specified.” The code K72.90 had excellent positive and negative predictive value for the development of HE in a prospective cohort of Child A and B cirrhosis [17]. The same code also successfully identified HE in a VA cohort meeting a validated definition of cirrhosis [17]. Several groups have used this ICD-10 code as one of many to identify cirrhosis; however, the specific performance of K72.90 in those algorithms is unknown [12,18,19].

Prescription data is accessible in many administrative databases. The treatment of HE is nearly uniform with one or two medications: lactulose and rifaximin. Consistency in HE treatment across different geographies and patient subgroups enhances the utility of prescription data in identifying the diagnosis. We found that a prescription for lactulose or rifaximin had high negative predictive value (0.99) and substantial positive predictive value (0.71) for HE [17].

Multiple gaps persist. Data are lacking regarding whether a given coding algorithm can identify patients with early stages of HE or whether diagnostic coding schema generalize between countries. Further, non-ICD-9 coding algorithms have only been validated in cohorts with known cirrhosis. These algorithms are not yet validated for use in larger, less-defined samples.

## 4. Administrative Databases: Which to Use

In Table 2, we detail the data elements and outcomes available in each dataset.

### 4.1. US Data

Many databases have been used to study the population burden and impact of HE. In the US, the lack of nationalized health care creates the central limitation of administrative data. Data for each patient is often dispersed across multiple payers and therefore databases. The Veterans Affairs (VA) data is rich and incudes diagnostic/procedure codes, laboratory data, and pharmacy records. However, even veterans receive care, both out- and inpatient at outside facilities with variable reconciliation of events and prescriptions. HE code and prescription-based algorithms have been validated using both ICD-9 and 10 [16,17].

The Organ Procurement and Transplant Network (OPTN) offers a database that includes all persons waitlisted for liver transplantation with granular data that is regulated by OPTN rules and manually entered by each transplant center. Among administrative data sources, OPTN data is unique given the richness of physiological variables and the intrinsic validity of the clinical diagnoses. A history of HE is recorded and HE is graded using the West-Haven scale.

The National Inpatient Sample (NIS) is an all-payer database of admission-level inpatient encounters strengthened by complete billing and in-hospital outcome data but lacking in laboratory and prescription information or data following discharge. The National Readmissions Database (NRD) is a sample of NIS data accounting for most states and hospitals contained within the NIS. In the NRD patients can be linked between hospitalizations by a unique identifier allowing for studies of readmissions albeit without accounting for the competing risk of post-discharge mortality. Although ICD-9 based algorithms for HE have been applied to these databases presuming similar performance compared to the VA, none have been validated [21].

Patients aged ≥65 years as well as those who are disabled or requiring hemodialysis are eligible for government insurance with Medicare. At a minimum, Medicare data includes longitudinal, patient level data linked to vital status records as well as comprehensive diagnosis/procedure codes and medications that are provided by a health care facility. The kind of research that can be performed using Medicare varies according to the data-elements at the investigator’s disposal. Algorithms using ICD-9 derived from the VA have been validated in Medicare data [22].

Finally, many investigators have used commercial claims data to study cirrhosis and HE-related outcomes [23,24]. Commercial claims vendors use highly varied data sources ranging from one sole insurer (Optum/United Health) or a pooled dataset from many employer-based insurance plans [25]. The richness of the claims data varies, some offer linkage to the originating provider-type while others do not, some offer laboratory claims but not the results of those laboratory tests. As such, careful inspection of the database’s data elements is necessary to understand the ability of each to capture the incidence, prevalence, burden, and outcomes of HE as well as the determinants thereof.

### 4.2. International Data

Canada has a universal health system, administered semi-independently by each of its 13 provinces and territories. The Institute for Clinical Evaluative Sciences holds administrative data for all Ontario residents utilizing publicly available insurance. Databases containing billing claims, hospitalization records, and death data are linked. Methods for identifying cirrhosis in claims data, validated in other cohorts, have been applied to this database [26]. Lapointe-Shaw and colleagues have validated the use of combinations of ICD-9 and 10 codes to identify cirrhosis and decompensated cirrhosis, but not HE specifically [12]. Outpatient physician claims for cirrhosis were sensitive but not specific, likely due to financial incentives provided for including a visit diagnosis of cirrhosis.

The National Patient Register contains diagnosis and hospital contact data on the entire population of Denmark since 1977. Diagnoses after 1994 were made using the ICD-10 coding system, and are notably entered by a physician, not other administrative personnel. Two studies have validated the use of ICD codes for cirrhosis in this registry [19,27], and numerous investigations into cirrhosis have been performed with it [5,28,29,30]. Jepsen and colleagues have reported on the incidence of HE in a cohort of alcohol-related cirrhosis from this registry, but the HE was identified by chart review and not administrative codes [5]. To date, no studies have validated the use of ICD-10 codes for HE in the Danish public registry.

Several studies of cirrhosis epidemiology have used a southern Swedish cohort, developed from a comprehensive population registry [31,32,33,34]. These studies initially identified 4611 patients with ICD-10 codes for cirrhosis, but 2950 were excluded by chart review as not meeting criteria for cirrhosis [31]. The authors identified the incidence of HE, defined as a prescription for lactulose. Another group recently used the Swedish National Patient Register, which collects ICD-10 codes for all specialty care in Sweden, and validated codes to identify cirrhosis [35]. No administrative codes have yet been used to identify HE within these cohort.

The European Liver Transplant Registry (ELTR), similar to UNOS in the United States, collects manually entered data regarding liver transplant indications and complications from 28 countries in Europe. While this registry includes pre-transplant data from patients with cirrhosis, there are no published studies of HE in this cohort.

## 5. Identifying Risk Factors for Hepatic Encephalopathy

Cohort studies aimed at identifying the incidence of new or interval HE will require patient samples with risk factors for HE development. Most studies have done this by identifying cirrhosis or a common cause of chronic liver disease, such as hepatitis C virus infection. As described above, algorithms for identifying cirrhosis have been validated in multiple datasets by multiple authors [7,8,9]. Coding algorithms have also been used to successfully identify cohorts with alcohol liver disease [8], non-alcoholic fatty liver disease [14,36], hepatitis C virus infection [8,37,38,39], and—with slightly less success—chronic hepatitis B virus infection [8,38,39]. Using the US Medicare database, we identified a cohort of patients with cirrhosis whose risk of incident diagnoses of HE were influenced by etiology (particularly alcohol-related liver disease), the presence of portal hypertension, comorbidities, and polypharmacy (particularly benzodiazepines, opioids, and proton pump inhibitors) [40]. The risk of HE was 11.6 per 100 person-years. Using the US VA database, we found that persons with cirrhosis and portal hypertension or an AST-to-Platelet Ratio Index >2.0 had a cumulative incidence in excess of 40% at 5 years. The specific risk factors identified included disease severity (albumin, total bilirubin), nonselective beta-blocker use, and statin therapy (inversely associated) [41].

## 6. Outcomes of HE

Several studies have used administrative data to describe the outcomes of persons with HE (Table 3). Scaglione demonstrated that HE was independently associated with mortality after hospitalization while Wong showed that grade of HE at the time of transplant evaluation was associated with increased mortality on the waitlist [24,42]. We showed using Medicare data that the median survival after HE was approximately 1 year for persons ≥65 years old as well as those with ascites prior to HE. In a claims database of privately insured persons, we found that the overall cumulative incidence of death at 1 year was 19% [25]. Stepanova and Hirode both examined the NIS and found that the in-hospital mortality and costs associated with hospitalizations for HE from 2005 to 2014 were approximately 17% and $17,000 [43,44]. Roggeri examined the global annual health care costs for Italian patients with HE and estimated approximately $15,000 USD [45].

## 7. Pitfalls of Administrative Data

There are three central limitations inherent to administrative data research: validity, completeness, and descriptive fidelity. First, we review, in Table 1, the database definitions of HE which have been validated. There are likely additional methods to identify patients with HE beyond this table. Codes such as ‘hepatitis C with coma (ICD-10 B19.21)’ or ‘encephalopathy (G93.41)’ may rarely be used to describe HE but we do not know their accuracy. Furthermore, the current method for identifying HE using ICD-10 codes requires pre-specifying a population with known liver disease. This method enhances data validity at the cost of inclusiveness. Using this method also makes the accuracy of identifying HE dependent on the techniques used to identify the liver disease population. Validated methods to identify HE without yet established cirrhosis coding are needed. Second, as reviewed in Table 2 and expanded upon above, each database varies with respect to its data elements or cross-sectional versus longitudinal design. Accuracy of diagnostic codes vary by population and database, possibly secondary to differences in reimbursement. Furthermore, in the context of disparate sources of health care funding, such as in the US, it can be unclear which portion of a given patient’s health care experience is captured within the dataset. Third, even valid and complete data may not be appropriate for specific aspects of HE care. No study, for example, has discerned the impact of covert from overt HE.

## 8. Future Directions

Future study should target two core areas: first, identify strategies to use multiple administrative data tools in tandem to identify patients who develop HE amongst those at risk; and second, linkage of administrative data to clinical care.

HE can be accurately identified by claims or prescription data, when done so within a cohort of known risk (i.e., HCV, cirrhosis). The next step is being able to expand these searches into larger population cohorts, by utilizing tools to first identify those at risk of HE. Natural language processing (NLP) holds potential future promise as an addition tool, beyond those discussed above, to identify patients with cirrhosis and risk of developing HE. NLP allow for automated extraction of text from medical charts, and could supplement administrative codes by also identifying “splenomegaly” or “varices” in radiology and endoscopic reports. An algorithm combining administrative codes and NLP of radiology report impressions had high (>90%) positive and negative predictive value for identifying cirrhosis [46]. A strategy that successfully uses multiple tools simultaneously including medications, laboratory values, codes, and NLP may optimally identify those at risk for HE from large databases.

Additional work must be done to leverage administrative data for clinical care. If hospital systems could efficiently and accurately identify patients at risk for the development of HE through administrative data, then those patients could be seamlessly incorporated into population health cohorts and targeted with additional resources. Given the availability of risk scores for HE using administrative data, these could be calculated and displayed at the point of care to influence decision making. If patients at hospital discharge could be automatically and accurately identified at high risk for recurrent HE, then linking those patients to close outpatient follow up and resources could optimize management. Finally, automated identification of patients at risk for HE with administrative data could facilitate clinical trial enrollment for studies aimed to treat this condition, and accelerate the pace of scientific discovery.

## 9. Conclusions

We cannot understand the societal burden of HE without administrative data. Rigorously collected data from prospective cohorts are essential tools for HE research. A research agenda that excludes the use of administrative data, however, does so at the peril of crucial insights. While each data stream is affected by its own pitfalls, those of administrative data are not intrinsically greater than conventional cohort studies. As reviewed, the tools required to avoid the pitfalls of administrative data are straightforward and readily available.

## Figures and Tables

**Table 1 jcm-09-03620-t001:** Methods to Identify Hepatic Encephalopathy Using Administrative Data.

Tool	Description	Study	Database	Relevant Result	Validated Method for Identifying HE or Cirrhosis	Limitations	Benefits
International Classification of Diseases, 9th Revision (ICD-9)	International standard for defining and reporting diseases 9th revision was used in the United States from 1979 to 2015 ICD-9 code for HE is 572.2	V. Lo Re et al. (2011)	Veterans Affairs	Nine of 295 patients with an ICD-9 code or laboratory value indicating liver dysfunction had an ICD-9 code for HE; the PPV of this code was 0.11 and estimated NPV of 0.99	HE	ICD-9 is not being coded in the United States after 2015, so available data ranges are limited; Variable accuracy in coding	International; Currently best validated; Specific code for HE
		Goldberg et al. (2012)	Local registry (two tertiary care centers)	Presence of one inpatient or outpatient ICD-9 code for cirrhosis, chronic liver disease, and a hepatic decompensation (of which HE was one), the PPV of 0.85 for confirmed cirrhosis	Cirrhosis		
		Kanwal et al. (2012)	Veterans Affairs	After identifying cirrhosis patients with ICD-9 codes and laboratory data, at least one ICD-9 code for HE had PPV of 0.86 and NPV of 0.87 for confirmed HE	HE		
		Nehra et al. (2013)	Local registry (single hospital system)	ICD-9 code for HE had PPV 0.92 and NPV 0.36 for identifying confirmed cirrhosis; did not report if it identified HE	Cirrhosis		
		Lapointe-Shaw et al. (2018)	Two Canadian hospitals	Having a single hospital diagnostic code for cirrhosis, including 572.2, was specific for cirrhosis (0.91–0.96 depending on subcohort), but not as sensitive (0.57–0.77); however, the authors did not specify in how many cases 572.2 was used vs. other codes	Cirrhosis		
International Classification of Diseases, 10th Revision (ICD-10)	United States began using ICD-10 in 2015 Many countries began using this system earlier No specific code for HE, instead many use K72.90	Thygesen et al. (2011)	Danish National Registry of Patients	The PPV of one inpatient or outpatient ICD-10 code for moderate/severe liver disease, which included K72.90, correctly identifying cirrhosis was 1.00; however, the authors did not specify in how many cases K72.90 was used vs. other codes	Cirrhosis	Only available in the United States 2015 and thereafter	International; Required to use data after 2015 in the United States; Readily available in most databases
		Mapakshi et al. (2018)	Veterans Affairs	Unable to validate the use of ICD-10 codes for HE because there were no HE events during the study period	Neither		
		Tapper et al. (2020)	Development cohort: single academic center Validation cohort: Veterans Affairs	In a validation cohort of veterans with HCV, ICD-10 code K72.90 identified development of HE with PPV 0.90 and NPV 0.93	HE		
		Lapointe-Shaw et al. (2018)	Two Canadian hospitals	Having a single hospital diagnostic code for cirrhosis, including K72.90, was specific for cirrhosis (0.91–0.96 depending on subcohort), but not as sensitive (0.57–0.77); however, the authors did not specify in how many cases K72.90 was used vs. other codes	Cirrhosis		
Prescription Data	Record of a medication prescription	Tapper et al. (2020)	Development cohort: single center Validation cohort: Veterans Affairs	In a validation cohort of veterans with HCV, lactulose prescription had PPV of 0.73 and NPV of 0.99 for HE diagnosis, while lactulose or rifaximin prescription had a PPV of 0.71 and NPV of 0.99	HE	Not available in every database	Lactulose therapy for overt HE is nearly uniform
Combination	ICD-9 + prescription data	Kaplan et al. (2015)	Veterans Affairs	An algorithm based on the ICD-9 code for HE and prescription fills for lactulose or rifaximin had weighted kappa agreement of 0.51 with the CTP-subscore for HE	HE	Not available in every database	Using multiple modalities in one algorithm can enhance predictive value

ICD, International Classification of Diseases; PPV, positive predictive value; NPV, negative predictive value; CTP, Child-Turcotte-Pugh.

**Table 2 jcm-09-03620-t002:** Potential Administrative Data Sources for Hepatic Encephalopathy Research.

Data Sources	Population	Data Elements	Outcomes	Validated Definition of Cirrhosis	Validated Definition of HE	Limitations
Veterans Affairs (VA)	National health care for US veterans	ICD-9/10CPT Physical exam Pharmacy Laboratory Imaging	Hospitalization Mortality Transplant Cost	Kanwal et al. (2012) V. Lo Re et al. (2011)	Kanwal et al. (2012) Kaplan et al. (2015) Tapper et al. (2020)	Male Missing outside data VA population and access to care may differ
Medicare	United States ≥65 years old	ICD-9/10CPT Pharmacy	Death Health care utilization Linked cohorts such as the Health and Retirement Study or Cardiovascular Health Study can provide additional outcomes relating to functional disability and cognitive function	Rakoski et al. (2012)	None	No laboratory data Relies on diagnosis and procedure codes
National Inpatient Sample (NIS) / National Readmissions Database (NRD)	United States Nationally representative sample All payers	ICD-9CPT	Length of stay Discharge disposition Inpatient mortality	None	None	No laboratory data available Relies on diagnosis and procedure codes alone and is subject to misclassification Inability to link hospitalizations to individual patients limits longitudinal follow-up post-discharge
Private Insurance Claims Data	United States Private insurance represents ~50% total market, often through employer	ICD-9CPT Pharmacy Laboratory	Hospitalization Direct health care costs Limited death data	None	None	Relies on diagnosis and procedure codes Enrolled only while covered Often missing death data
National Patient Registries	Denmark, Sweden, Ontario	Includes detailed information on clinical characteristics, laboratory data, imaging, procedures and outcomes	Hospitalization Death Additional data depending on registry	Thygesen et al. (2011) Lapointe-Shaw et al. (2018)	None	Country and health care system specific
Organ Procurement and Transplant Network (OPTN)	United States Listed for liver transplantation	Manually entered detailed pre-, intra-, and post-transplant clinical information	Data on liver transplantation, and post-liver transplant outcomes Linked by UNOS to social security death index	None (manually input by transplant program)	None (manually input by transplant program)	Considerable selection bias given limited to transplant centers and listed patients Potential for misclassification due to inaccurate completion of questionnaire ELTR: No information on patient ethnicity or socioeconomic information
European Liver Transplant Registry (ELTR)	Europe (155 centers from 28 countries)	Detailed information on liver transplant indications, transplant types and complications	Death Transplant outcomes	None (manually input by transplant program)	None	

Some elements of this table were adapted from Moon et al. (2019) [20].

**Table 3 jcm-09-03620-t003:** Administrative Studies Detailing the Outcomes Associated with Hepatic Encephalopathy (HE).

	Study	Population	Definition of HE	Outcome(s)
Incidence/Prevalence	Tapper	US Veterans with APRI>2.0 2005–2015	ICD-9 572.2 or the use of lactulose and/or rifaximin	The cumulative probabilities of overt HE at 1, 3, and 5 years was 22.6%, 36.9%, and 43.6%
	Tapper	US Medicare 2008–2015		Incidence rate: 11.6 per 100 person-years
	Nilsson	Sweden, 43% with ascites	Lactulose use	Cumulative incidence at 1 and 10 years, 6.4% and 26%
Mortality	Wong	Transplant waitlisted Americans 2003–2012	Manually entered grading	HE is associated with mortality: Grade 1–2 1.1.3 (1.02–1.26) Grade 3–4: 1.65 (1.44–1.89)
	Scaglione	Privately insured Americans with cirrhosis and a readmission 2010–2014	572.2	Adjusted mortality associated with HE 1.14 (1.04–1.24)
	Tapper	US Medicare 2008–2015 Optum commercial claims 2008–2015	ICD-9 572.2 or the use of lactulose and/or rifaximin	Median survival 0.95 and 2.5 years for those ≥65 or <65 years old; 1.1 and 3.9 years for those with or without ascites
Post-transplant mortality	Wong	Transplant waitlisted Americans 2003–2013	Manually entered grading	HE is associated with mortality: Grade 3–4: 1.27 (1.17–1.39)
Inpatient outcomes	Hirode	Hospitalized Americans 2010–2014	ICD-9 572.2	In-hospital mortality 12.3% from 13.4% Cost per admission 16,168 to 16,919
	Stepanova	Hospitalized Americans 2005–2009	ICD-9 572.2	In-hospital mortality 15.6% to 14.3% Cost per admissions 16,512 to 17,812
	Tapper	US Medicare 2008–2015 Optum commercial claims 2008–2015	ICD-9 572.2 or the use of lactulose and/or rifaximin	11.8 (IQR 2.9–38.0) hospital days per person-year Combination lactulose and rifaxmin use associated with lower hospital days and 30 day readmission
Costs	Roggeri	Hospitalized Italians 2011	ICD-9 572.2	Annual HE costs: 15,295 USD

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
