# Peer review of "The Use of Administrative Data to Investigate the Population Burden of Hepatic Encephalopathy"

_jcm, 2020, doi:10.3390/jcm9113620_

Round 1

Reviewer 1 Report

Dear Editors!

First, let me thank you for the opportunity to review the paper by Bloom and Elliot. It is a comprehensive and well written paper on methodology problems regarding the register-based studies on hepatic enchephalopaty. It point out several important aspects of previous attempts to evaluate the incidence, risk factors and burdens of enchephalopathy.

Second, some broad comments. The paper starts with an introduction where it states that the increased health-care utilization by cirrhosis is driven by the cirrhosis-related complications such as encephalopathy. The rest of the paper do explore the difficulties of studying that complication. I do not think that the references (5,6) supports the sentence.

In section 2 the last paragraph evaluate the use of data set to identify the diagnosis cirrhosis. More specifics on whether any of the attempts have been successful should be stated.

In section 4 the 4.1 section o US databases are long and detailed with much information that should be well-known to the reader. In 4.2 the Danish and Swedish registers are described and in the table Finland, Iceland and Norway are included, but they are not mentioned in the text. The Swedish study (35) does only study "alcohol cirrhosis K70.3" and the diagnosis "cirrhosis K74.6", and it does not explore the other diagnoses of chronic liver disease that leads to cirrhosis. This is one of the difficulties in diagnosing cirrhosis that the authors do not explore in the text. PPV and NPV will of course be good if one only use "alcohol cirrhosis", but all patients with hemochromatosis, viral hepatitis,autoimmune hepatitis, primary biliary cholangitis and sclerosing cholangitis with cirrhosis will not be included.

In addition, the authors do not discuss whether changes in reimbursment for the diagnosis encephlopathy or cirrhosis may affect the use of the diagnosis.

Third - one small specific comment: on line 79 there's a this too much.

Reviewer 2 Report

Thank you for the opportunity to review this manuscript by Bloom and Tapper which is a systematic review of the ability of administrative healthcare data to identify the presence of hepatic encephalopathy. This is an important topic given the growing burden of cirrhosis and associated complications in the United States and the increased interest and ability to use large administrative databases to perform health services research.

The authors have submitted a well-written, through, and encompassing review on the strengths and weakness of using ICD coding to define HE in databases around the world. 

This review is unique in that a similar review has never been published therefore, it is an important contribution to the literature and I anticipate it will be well cited once published.

I have no specific edits or comments to suggest at this time.

Author Response

n/a